# Kinetics of Chromium Reduction Associated with Varying Characteristics of Agricultural Soils

Chia-Yu Yang [1], Yan-Lin Tseng [2] and Zeng-Yei Hseu [1,*]

1 Department of Agricultural Chemistry, National Taiwan University, Taipei 10617, Taiwan; d10623003@ntu.edu.tw
2 Department of Environmental Science and Engineering, National Pingtung University of Science and Technology, Pingtung 91201, Taiwan; garysul500hp@gmail.com
* Correspondence: zyhseu@ntu.edu.tw; Tel.: +886-927-613-668

**Abstract:** Chromium (Cr)(VI) is carcinogenic; thus, the excessive presence of Cr(VI) in soils can pose potential risks to water quality, food safety, and human health. The kinetics of Cr(VI) reduction in soils are important for assessing the fate of Cr in the environment. The present study tested physio-chemical and microbial properties in twenty-eight agricultural soils collected in Taiwan to evaluate the relationship between the reduction rate of Cr(VI) and soil properties, using 49-day incubation at 25 °C. At the beginning of incubation, 100 mg Cr(VI) $kg^{-1}$ was spiked into the soils. The reduction of Cr(VI) was described by first-order kinetics at a significant level ($p < 0.05$) for the tested soils. The rate constant (k) of Cr(VI) reduction ranged from 0.01 to 4.21 $day^{-1}$. In addition, the k value significantly increased with organic carbon (OC) and cation exchange capacity, but significantly decreased with increasing pH and dithionite-citrate-bicarbonate extractable Mn ($Mn_d$). However, a predictive model using stepwise regression analysis indicated that the k value of the kinetics was controlled by OC, dissolved organic carbon (DOC), and $Mn_d$, thereby identifying the complex interactions between Cr(VI) reduction and soil factors in the humid tropics.

**Keywords:** first-order kinetics; heavy metal; soil contamination; speciation of chromium





## 1. Introduction

Chromium, which is one of the most toxic heavy metals in the environment, occurs at levels of 10 to 100 mg $kg^{-1}$ in natural soils depending on different parent materials, but the concentration of Cr in soils can reach up to 4000 mg $kg^{-1}$ due to human activities [1,2]. The wide applications of Cr in numerous industrial activities such as electroplating, leather tanning, Cr-ore residue processing, and the weathering of Cr-rich rocks are potential sources of Cr in soils [3–5]. Cr exists in different oxidation states, ranging from +2 to +6 naturally, while the most stable forms are Cr(III) and Cr(VI) [6]. Cr(VI), which is known to be carcinogenic, has much higher solubility and toxicity than Cr(III) in the environment [1]. The Cr released into soils can be oxidized to Cr(VI), which is easily leached out, causing the contamination of ground and surface water [7]. To remove Cr(VI) in the environment, the major chemical strategies include reduction, adsorption, precipitation, and ion exchange [8]. Among these remediation strategies, reduction of Cr(VI) has received significant attention during the past few decades [9,10].

The reduction of Cr(VI) to Cr(III) is an essential reaction in determining the mobility and toxicity of Cr in soils [11]. Commonly, a portion of Cr(VI) released into soils may be reduced immediately by reducing agents, while the rest of the Cr(VI) may remain in the environment for a long time [12]. However, this process is complicated because of the mutual interaction of Cr and various soil properties, including physio-chemical and microbial effects. For example, soil organic matter is a major reductant of Cr(VI), as it serves as an electron acceptor and reduces Cr(VI) by becoming $CO_2$ [13,14]. Conversely, manganese (Mn) oxide is known for promoting the oxidation of Cr(III) in soils though

active oxidizing sites on the mineral's surface [15]. Unlike Mn oxides, iron (Fe) oxides retain Cr(III) to form (Cr, Fe) (OH)$_3$, limiting the oxidation of Cr(III) in soils [16]. Besides these soil components, other soil properties also influence the reduction of Cr(VI). For example, low soil pH enhances the reduction of Cr(VI) [17]. However, the reduction rate of Cr(VI) decreases with the increase of soil clay and cation exchange capacity (CEC), because of the high potential of Cr retention in soils [18]. From the biotic aspect, the microbial reduction of Cr(VI) can occur either directly, through enzyme reaction, or indirectly, thorough a bacterially-produced reducing agent [5]. Microbial reduction is controlled by microbial activity, which is highly affected by soil properties such as pH, temperature, and the amount of carbon in soils [19]. In addition, microbial biomass is another decisive factor for microbial reduction of Cr(VI) [19].

The added Cr(VI) in soils may be reduced by electron donors as mentioned and then adsorbed or precipitated as Cr(III) [12]. To understand this complicated process, the kinetics of Cr(VI) reduction as a function of process time have been used to describe the fate of Cr affected by different soil properties. Xiao et al. investigated the reduction of Cr(VI) and found that the processes are successfully described by first-order kinetics, and the kinetic rate constant varied from 0.01 day$^{-1}$ in the incipient soil (Inceptisol) to 0.11 day$^{-1}$ in the highly-weathered soil (Oxisol) [20]. Xiao et al. further found that the rate constant significantly increased with the content of ferrous ion, clay, and the richness of the microbial population [20]. Yang et al. found that kinetic of Cr(VI) reduction followed the first-order reaction in soils containing an initial concentration of 187.5 mg Cr(VI) kg$^{-1}$ [21]. They also showed that the rate constant was controlled by the soil's specific surface area and pH. Additionally, the rate constant is also sensitive to soil CEC and Mn oxides [15,18]. These studies mainly used soils from temperate regions; therefore, information on Cr(VI) reduction is still insufficient in humid tropical soils.

The kinetics of Cr(VI) reduction in soils must consider multiple soil properties, as mentioned. Therefore, we hypothesized that the kinetics of Cr(VI) reduction are controlled by the soil's physical, chemical, and microbial properties. In this study, we collected different soil samples covering representative soil types in agricultural lands in Taiwan to (i) examine the kinetics of Cr(VI) reduction in the soils by first-order reaction and (ii) elucidate the relationships between the reduction rate of Cr(VI) and soil properties to predict the kinetics of Cr(VI) reduction in humid tropical soils.

## 2. Materials and Methods

### 2.1. Collection and Characterization of Soil Samples

Twenty-eight soil samples were collected from 22 soil series on agricultural lands through Taiwan, including 20 surface soils and eight subsurface soils. In the category of Soil Order based on the U.S. soil taxonomy [22], these soils were classified as Andisol, Inceptisols, Alfisols, Vertisols, and Ultisols, which are the dominant soil types of agricultural lands in Taiwan [23]. All tested samples were dried, ground, and passed through 10 mesh sieves prior to laboratory analysis.

Soil particle size distribution was determined via the pipette method [24]. Soil pH was measured in a mixture of soil and deionized water (1:1, *w/v*) using a glass electrode [25]. Total organic carbon (OC) content was determined using the Walkley–Black wet oxidation method [26]. The CEC was determined via the ammonium acetate method (pH 7.0) [27]. Soils were extracted by the dithionite-citrate-bicarbonate (DCB) extraction method to obtain the amount of crystalline and non-crystalline oxides of Fe (Fe$_d$) and DCB-extractable Mn (Mn$_d$) [28]. Soil samples were extracted by NH$_4$OAc–hydroquinone to analyze the easily-reducible Mn (Mn$_r$) [29]. All metal contents in the solutions were measured by atom absorption spectroscopy (AAS) (Z-2300, Hitachi, Tokyo, Japan). The analysis of total Cr(VI) content was conducted with the alkaline digestion method, and the concentration of Cr(VI) in the digest was determined colorimetrically at a wavelength of 540 nm using a 1,5-diphenylcarbazide (DPC) solution [30].

The soil's dissolved organic carbon (DOC) was extracted using the method designed by Jones and Willett [31]. The extracted solution was then analyzed with a total organic carbon analyzer (Shimadzu TOC-VCHS, Tokyo, Japan). The microbial biomass C and N were determined using the chloroform fumigation–direct extraction technique [32,33]. The fresh-soil sample was extracted with 0.5 mol $K_2SO_4$, and organic carbon in the extract was determined by the Walkley–Black wet oxidation method [26], whilst the microbial biomass N was determined using the Kjeldahl method by distillation.

### 2.2. Incubation Experiments

Two hundred grams of the soil sample was placed into a 400 mL plastic can. Then, $K_2Cr_2O_7$ was added to the soil to meet the concentration of 100 mg Cr(VI) $kg^{-1}$. After thoroughly mixing the soil, the moisture of the soil sample was kept at 60% of the water holding capacity and the soil was incubated at 25 °C for 49 days. The exchangeable Cr(VI) in the soil was measured at day 1, 7, 14, 21, 28, 35, 42, and 49 after the addition of Cr(VI).

The exchangeable Cr(VI) concentration during incubation was analyzed using the method proposed by Bartlett [12]. Five grams of the soil sample were extracted with 15 mL 0.01 mol $K_2HPO_4/KH_2PO_4$ (pH 7.2) and the extracted Cr(VI) was determined colorimetrically by the DPC solution, as mentioned above.

### 2.3. Calculation and Statistical Analysis

Since first-order kinetics provide a simple and straightforward equation as a function of time, the exchangeable Cr(VI) over time during incubation was described by a first-order kinetic model: $\ln[A] = \ln[A_0] - kt$, where A is the concentration (mg $kg^{-1}$) of Cr(VI) at time (t), $A_0$ is the initial concentration (mg $kg^{-1}$) of Cr(VI), and k is the rate constant ($day^{-1}$). A linear correlation matrix of the k value and all soil properties was determined using Pearson correlation at a significant level of 0.05. Further evaluation of stepwise multiple linear regression analysis was used to extract the significant soil variables and to generate a predictive model for the reduction rate of Cr(VI) in soils. All statistical analyses were done using R (4.1.2 version).

## 3. Results and Discussion

### 3.1. Soil Characteristics

The total Cr in the studied soils was variable and ranged from 15.8 to 1520 mg $kg^{-1}$ (Table 1), but the mean was slightly higher than the globally-averaged concentration (100 mg $kg^{-1}$) [34]. Moreover, the concentration of total Cr(VI) was much lower than the total Cr in all soils, which ranged from 1.34 to 29.4 mg $kg^{-1}$. The soil pH ranged from 4.0 to 8.1. The lowest value of the OC content was 2.64 g $kg^{-1}$ and the highest value was up to 150 g $kg^{-1}$. On averagely, the OC content was 18.9 g $kg^{-1}$ in studied soils. The DOC of all soil samples ranged from 3.61 to 47.4 mg $kg^{-1}$. The lowest value for CEC among soils was 3.62 $cmol_{(+)}$ $kg^{-1}$, while the highest was 48.2 $cmol_{(+)}$ $kg^{-1}$. The sand fraction ranged from 2 to 79%. The clay content ranged from 15 to 80%. The $Fe_d$ content varied greatly in the studied soils, ranging from 1.22 to 43.8 g $kg^{-1}$, whereas the $Mn_d$ content was much lower than the $Fe_d$ content, ranging from 0.03 to 0.87 g $kg^{-1}$. Additionally, the lowest $Mn_r$ content was 2.00 mg $kg^{-1}$, while the highest was 355 mg $kg^{-1}$. The lowest MBC content was 0.02 g C $kg^{-1}$ and the highest was 3.64 g C $kg^{-1}$. The lowest MBN content was 0.08 g N $kg^{-1}$ and the highest was 1.39 g N $kg^{-1}$.

**Table 1.** Soil properties of the studied soils (*n* = 28).

| Item | Mean | SD | Min. | Max. |
|---|---|---|---|---|
| Total Cr (mg kg$^{-1}$) | 124 | 293 | 15.8 | 1520 |
| Total Cr(VI) (mg kg$^{-1}$) | 8.66 | 8.48 | 1.34 | 29.4 |
| pH | 6.22 | 1.24 | 4.00 | 8.10 |
| OC (g kg$^{-1}$) | 18.9 | 26.2 | 2.64 | 150 |
| DOC (mg kg$^{-1}$) | 10.6 | 9.36 | 3.61 | 47.4 |
| CEC (cmol$_{(+)}$ kg$^{-1}$) | 14.4 | 10.6 | 3.62 | 48.2 |
| Fe$_d$ (g kg$^{-1}$) | 16.4 | 9.72 | 1.22 | 43.8 |
| Mn$_d$ (g kg$^{-1}$) | 0.30 | 0.17 | 0.03 | 0.87 |
| Mn$_r$ (mg kg$^{-1}$) | 87.3 | 80.4 | 2.00 | 355 |
| Sand (%) | 23.7 | 19.5 | 2.00 | 79.0 |
| Silt (%) | 34.1 | 14.0 | 6.00 | 59.0 |
| Clay (%) | 42.2 | 15.3 | 15.0 | 80.0 |
| MBC (g C kg$^{-1}$) | 1.02 | 0.87 | 0.02 | 3.64 |
| MBN (g N kg$^{-1}$) | 0.65 | 0.36 | 0.08 | 1.39 |

*3.2. Reduction Processes of Chromium (VI) in the Studied Soils*

To present the diverse reduction processes of Cr(VI) in the studied soils, we selected six tested soils (Soil 8, Soil 11, Soil 12, Soil 19, Soil 24, and Soil 26) to illustrate the concentration of exchangeable Cr(VI) over time in Figure 1. In day 1, the Cr(VI) rapidly decreased, ranging from 1.48 mg kg$^{-1}$ in the soil with andic properties (Soil 24) to 64.1 mg kg$^{-1}$ in the highly-weathered soil (Soil 12). The reduction of Cr(VI) in the studied soils occurred in the beginning of incubation, which was consistent with the report by Bartlett and James [12]. Afterward, the exchangeable Cr(VI) in these soils gradually declined in the later incubation period, indicating that the electron donors were almost exhausted and the reduction of Cr(VI) became slow. At the end of the incubation, the exchangeable Cr(VI) was 35.0 mg kg$^{-1}$ in Soil 08, 2.72 mg kg$^{-1}$ in Soil 11, 43.7 mg kg$^{-1}$ in Soil 12, 10.0 mg kg$^{-1}$ in Soil 19, 1.67 mg kg$^{-1}$ in Soil 24, and 0.30 mg kg$^{-1}$ in Soil 26 (Figure 1). A slight increase of Cr(VI) in some soil samples may have resulted from the oxidation of soluble Cr(III) in the initial soils [11]. The level of Cr(VI) remaining in the soils after 49 days ranged from 0.30 to 43.7% of the initial Cr(VI), revealing a large variation of Cr(VI) reduction in the soils.

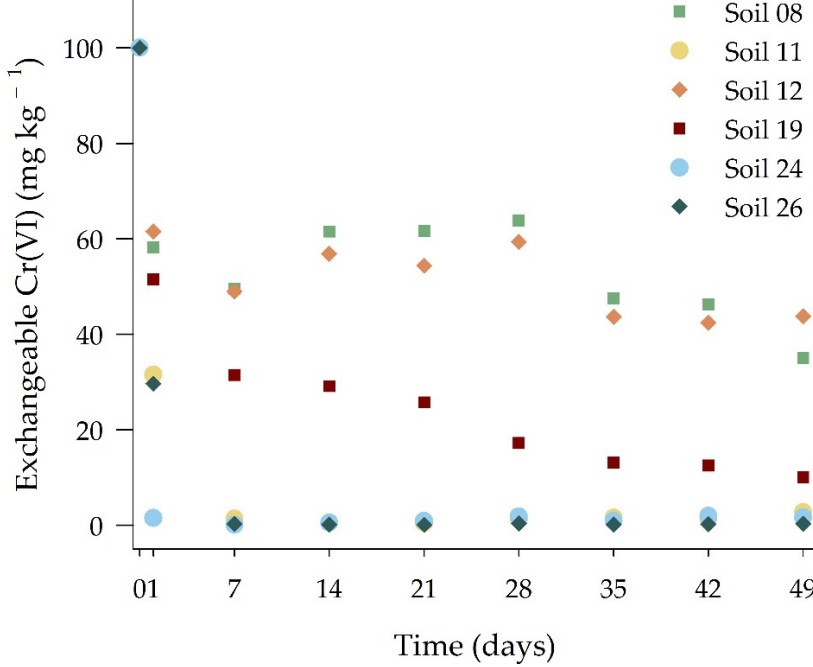

**Figure 1.** Time courses for Cr(VI) with an initial concentration of 100 mg Cr(VI) kg$^{-1}$ in selected soils.

The reduction rate of Cr(VI) was highest in Soil 24, while the lowest value presented in the subsoil of strongly weathered soil (Soil 08) (Figure 1). Soil 24 was classified as Andisol and was rich in OM, which accelerated the reduction of Cr(VI). Additionally, Soil 24 exhibited a high level of CEC, providing stable adsorption of Cr, so the exchangeable Cr(VI) detected in the soil was lower than in other soils [18]. The low content of $Mn_d$ and $Mn_r$ further suggests low oxidation capacity in the soils [12]. On the contrary, Soil 08 exhibited the least DOM and MBN, implying that it had a low reducing power for Cr(VI) due to the lack of reductant and chromium-reducing bacteria in soils [19].

The rate constant of Cr(VI) reduction (k) ranged from 0.01 to 4.21 $day^{-1}$, with a mean of 0.67 $day^{-1}$ (Table 2). All $R^2$ values were higher than 0.45, suggesting that the Cr(VI) reduction followed first-order kinetics in the soils with $p$ values below 0.05. Moreover, the k values in most cases of the present study were much higher than those reported in the temperate regions [20,21]. Most of the soils with low k values (<0.05 $day^{-1}$) were subsoils derived from sandstones, slates, serpentinites, or quaternary alluvium (Soil 02, 03, 06, 08, 12, 23, 28) (Table 2), which suggests that parent materials did not directly influence the reduction of Cr. Moreover, the contents of OM and DOM of these subsoils were below the average levels of all soils, while the other properties seemed to have no consistent trend. Particularly, pH values ranged from 4.0 to 8.1, and particle-size distribution varied greatly as well. Soils 17, 18, 19, 20, and 22 had higher reduction rates of Cr, with k values of 0.06 to 0.62 $day^{-1}$. These soils were characterized by higher pH (7.2 to 7.9) and lower CEC and $Fe_d$ content than the mean levels in the studied soils. The last portion of the samples exhibited higher reduction rates in which the k values were 0.71 to 4.21 $day^{-1}$. The pHs of these soils were all under 7, except for the soil 21 (pH = 7.9), whereas the other soil properties were quite diverse.

**Table 2.** Rate constant (k) of Cr(VI) reduction in all soils (*n* = 28).

| Soil Code | Reduction Rate Constant, k | $R^2$ | *p* Value |
|---|---|---|---|
| 01 | 1.02 | 0.99 | <0.0001 |
| 02 | 0.03 | 0.63 | 0.0215 |
| 03 | 0.04 | 0.61 | 0.0289 |
| 04 | 1.30 | 1.00 | <0.0001 |
| 05 | 1.04 | 0.99 | <0.0001 |
| 06 | 0.02 | 0.50 | 0.0404 |
| 07 | 0.80 | 0.98 | 0.0005 |
| 08 | 0.01 | 0.47 | 0.0497 |
| 09 | 1.07 | 0.99 | <0.0001 |
| 10 | 0.78 | 1.00 | <0.0001 |
| 11 | 1.15 | 1.00 | <0.0001 |
| 12 | 0.01 | 0.49 | 0.0380 |
| 13 | 0.02 | 0.51 | 0.0343 |
| 14 | 0.03 | 0.60 | 0.0222 |
| 15 | 0.71 | 0.92 | 0.0167 |
| 16 | 1.24 | 1.00 | <0.0001 |
| 17 | 0.13 | 0.83 | 0.0203 |
| 18 | 0.37 | 0.97 | <0.0001 |
| 19 | 0.06 | 0.78 | 0.0119 |
| 20 | 0.62 | 1.00 | <0.0001 |
| 21 | 0.73 | 1.00 | <0.0001 |
| 22 | 0.07 | 0.85 | 0.0080 |
| 23 | 0.01 | 0.47 | 0.0436 |
| 24 | 4.21 | 1.00 | 0.0018 |
| 25 | 0.82 | 0.95 | 0.0042 |
| 26 | 1.22 | 1.00 | <0.0001 |
| 27 | 0.79 | 0.97 | 0.0008 |
| 28 | 0.02 | 0.68 | 0.0091 |

### 3.3. The Relationship between Hexavalent Chromium Reduction and Soil Properties

To elucidate the effects of general soil properties on the reduction rate of Cr(VI), a linear correlation was conducted with the partial dataset in Table 1; the correlation is listed in Table 3. The k value positively correlated with OC and CEC at significant levels of 0.001 and 0.01, respectively, whereas pH and $Mn_d$ had significantly negative relationships with the k value ($p < 0.05$). Nevertheless, DOC, $Fe_d$, $Mn_r$, sand, silt, clay, MBC, and MBN correlated poorly with the k value. The k value significantly and negatively correlated with soil pH, suggesting that the reduction rate of Cr(VI) was higher in acidic soils due to the prevalent hydrogen ions, which enhanced the reduction of Cr(VI) back to Cr(III) (Table 3) [17,35]. Moreover, the pH was strongly related to CEC, $Fe_d$, silt, and clay in the soils, which indirectly promoted the reduction of Cr(VI) as well [16,18]. Furthermore, $Mn_d$ was negatively correlated to the k value, implying that $Mn_d$ enhanced the oxidation of Cr(III) and increased the amount of Cr(VI) in the soils [12]. In contrast, OC and CEC significantly increased the k value. The reduction of Cr(VI) is enhanced by OC in the environment [13]. Additionally, CEC was related to the retention of Cr(III), preventing Cr from oxidizing [18]; thus the k value was high in the soil with high CEC.

**Table 3.** Pearson coefficients of linear correlation between the rate constant (k) of Cr(VI) reduction and soil properties.

| Item | pH | OM | DOC | CEC | $Fe_d$ | $Mn_d$ | $Mn_r$ | Sand | Silt | Clay | MBC | MBN |
|---|---|---|---|---|---|---|---|---|---|---|---|---|
| OM | −0.27 | | | | | | | | | | | |
| DOC | −0.13 | −0.05 | | | | | | | | | | |
| CEC | −0.38 * | 0.67 *** | 0.16 | | | | | | | | | |
| $Fe_d$ | −0.61 *** | 0.01 | −0.07 | 0.46 * | | | | | | | | |
| $Mn_d$ | −0.01 | −0.25 | −0.19 | 0.17 | 0.38 * | | | | | | | |
| $Mn_r$ | 0.32 | −0.14 | −0.05 | 0.29 | 0.12 | 0.82 *** | | | | | | |
| Sand | 0.10 | −0.07 | 0.15 | −0.27 | −0.58 ** | −0.31 | −0.19 | | | | | |
| Silt | 0.41 * | 0.25 | −0.14 | 0.11 | −0.08 | 0.23 | 0.38 * | −0.63 *** | | | | |
| Clay | −0.50 ** | −0.14 | −0.06 | 0.23 | 0.81 *** | 0.18 | −0.10 | −0.70 *** | −0.11 | | | |
| MBC | −0.12 | −0.19 | −0.05 | −0.21 | 0.16 | −0.17 | −0.14 | 0.17 | −0.04 | −0.18 | | |
| MBN | −0.24 | −0.21 | 0.37 | −0.04 | 0.65 *** | −0.35 | −0.42 * | −0.25 | −0.14 | 0.47 * | 0.21 | |
| k | −0.38 * | 0.85 *** | 0.29 | 0.58 ** | −0.04 | −0.44 * | −0.29 | 0.08 | 0.10 | −0.19 | 0.12 | 0.07 |

\* $p < 0.05$, \*\* $p < 0.01$, \*\*\* $p < 0.001$.

To further predict the reduction rate of Cr(VI) in the soils, a stepwise regression was used by applying the data set in Table 3. The stepwise regression model is described below:

$$k = 0.158 + 0.015\,OC + 0.027\,DOC − 0.939\,Mn_d \quad (R^2 = 0.87, p < 0.01) \tag{1}$$

The soil properties involved in the model were OC, DOC, and $Mn_d$. Table 4 lists the summary of these variables. Soil OC was the major factor to control the reduction rate of Cr(VI) and explained 73% of the variance in the predicting model. The result was in accordance to the significantly positive relationship between OC and the k value. DOC was not significantly correlated with the k value. However, the effect of DOC cannot be neglected, as the DOC represents the labile part of organic matters in soils [14]. Furthermore, $Mn_d$ was related to the k value at a significant level, and thus was also identified as an important factor in the model. Overall, this model describes 87% of the variance in Cr(VI) reduction, suggesting that the model is successful in predicting the reduction rate of Cr(VI) in the studied soils.

**Table 4.** Summary of stepwise regression model for predicting the k value based on soil properties.

| Parameter | Parameter Estimate | *p* Value | Partial $R^2$ | Model $R^2$ |
|---|---|---|---|---|
| Intercept | 0.158 | 0.382 | | |
| OC | 0.015 | 0.000 | 0.73 | 0.73 |
| DOC | 0.027 | 0.000 | 0.11 | 0.84 |
| $Mn_d$ | −0.939 | 0.024 | 0.03 | 0.87 |

Stepwise regression analysis was done with the variable selection method, with entering and removing of parameters at $p < 0.05$ and $p > 0.10$, respectively.

By using the stepwise regression model, Xiao et al. deduced that the reduction of Cr(VI) was controlled by DOC, Fe (II), pH, and clay [20]. A significant correlation between the k value and pH was exhibited in this study. However, the model in this study demonstrated that the behavior of Cr(VI) reduction in soils from humid tropics was controlled by OC, DOC, and $Mn_d$, regardless of the pH and clay in the present study.

Table 4 shows that the k value was significantly correlated with pH, OC, CEC, and $Mn_d$, while the particle size distribution and the microbial biomass C and N were not significantly related. However, the stepwise regression model implied that the kinetics of Cr(VI) reduction were predicted by OC, DOC, and $Mn_d$, which accounted for 87% of the variance. The predictive model of Cr(VI) reduction in the soils with these properties is helpful to evaluate the dynamics of Cr oxidation and reduction in soils of the humid tropics.

## 4. Conclusions

Based on the variation of exchangeable Cr(VI) in the soils during the 49-day incubation, the kinetics of Cr(VI) reduction were successfully described by first-order reaction at a significant level ($p < 0.05$). The reduction rate constant k calculated by the kinetic equation ranged from 0.01 to 4.21 $d^{-1}$ in the studied soils. In addition, the k value significantly increased with OC and CEC, but decreased with pH and $Mn_d$. However, a significant predictive model generated by stepwise regression analysis indicated that the kinetics of Cr(VI) reduction were controlled by OC, DOC, and $Mn_d$. By using this model, we demonstrated different behaviors of Cr(VI) reduction and achieved a quantitative prediction of Cr(VI) reduction rates in agricultural soils in humid tropics.

**Author Contributions:** Experiments, data curation, formal analysis, writing—original draft preparation, and visualization, C.-Y.Y.; experiments, Y.-L.T.; conceptualization, writing—review and editing, and supervision Z.-Y.H. All authors have read and agreed to the published version of the manuscript.

**Funding:** This research was financially supported under Contract No. MOST 108-2313-B-002-040-MY3 by the Ministry of Science and Technology, Taiwan.

**Institutional Review Board Statement:** Not applicable.

**Informed Consent Statement:** Not applicable.

**Data Availability Statement:** Data used in this study are duly available from the first authors on reasonable request.

**Acknowledgments:** The authors are thankful to Ying-Hsiu Chang for collecting the studied soil samples.

**Conflicts of Interest:** The authors declare no conflict of interest.

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
