# Peer review of "Kinetics of Chromium Reduction Associated with Varying Characteristics of Agricultural Soils"

_water, doi:10.3390/w14040570_

Round 1

Reviewer 1 Report

The authors write about their experience in investigating and tested physio-chemical and microbial properties in 12 twenty-eight agricultural soils collected in Taiwan to evaluate the relationships between the reduc- 13 tion rate of Cr(VI) and soil properties by 49-day incubation at 25°C. The manuscript has a proper structure and in general it is well written.Although the study is interesting and could be useful for a certain group of scientific fartenity, therefore, I would suggest improving the manuscript substantially, giving a chance for the next round.

-The introduction is not cohesive. It is just a compilation of information gathered from literatures. Please rewrite.

- The sampling design described here is very short. Please give more details

-Line 10. Chromium (Cr) (VI) is carcinogenic and can cause potential risks in soils to water quality, food safety, and human health. Please rewrite

-Line 13. twenty-eight agricultural soils collected. Soils types or soil samples?. And line 59 representative soil types. Why selected different soils types as Andisol, Inceptisols, Alfisols, Vertisols, and Ultisols, in line 65.

-Line 61. Yang et al. found. Reference number?

-line 64. Twenty-eight soil samples were collected from surface or subsurface layers of 22 agricultural lands. Indicate which are superficial and which are not.

Author Response

Dear reviewer:

Reviewer 2 Report

Manuscript entitled “Kinetics of Chromium Reduction Associated with Varying Characteristics of Agricultural Soils” submitted by Chia-Yu Yang, Yan-Lin Tseng and Zeng-Yei Hseu, can be considered for publication in Water Journal, after a serious major revision.

Here is a list of my specific comments:

  1. Page 1, 1. Introduction: This section is quite brief and should be detailed. The most important aspects related to this topic should be presented and discussed, in order to provide a clear description of the state of art in this field. Also, at the end of Introduction, the main objectives of this study should be detailed presented.
  2. Page 2, 2.1. Collection of soil samples: (a) This title should be deleted, and the first two and the last paragraphs should be included in the next section. (b) The paragraphs “The climate in Taiwan…rainfall and high air temperature.” should be deleted, because are irrelevant for this discussion.
  3. Page 2, 2.2. Characterization of soil samples: In this section provide a clear presentation of the experimental methodology, the experimental parameters examined in this study, and the analytical methods used for this.
  4. Page 3, line 102: “…was described by a first-order kinetic model…”. Why only this model???
  5. Page 3, 3. Results and Discussion: This section should be reorganized. The experimental results included in this section should be discussed in detail, in accordance with the main objectives of this study. Only their presentation is not enough for a scientific paper.
  6. Page 4, Table 2: This table should be transformed in a figure.
  7. Page 6, 4. Conclusions: This section is too brief and should be detailed. Include here the most important experimental results and findings to highlight the importance of this study.

Author Response

Dear reviewer:

Round 2

Reviewer 1 Report

The authors have done a great job and have considered the reviewers' indications. I think the work can be published 

Reviewer 2 Report

All my previous remarks and comments have been considered into new version of the manuscript. It means that revised manuscript meets the criteria, and in my opinion can be published as technical report in Water Journal.